# Interactive Effect of Biological Agents Chitosan, Lentinan and Ningnanmycin on Papaya Ringspot Virus Resistance in Papaya (*Carica papaya* L.)

**DOI:** 10.3390/molecules27217474

**Published:** 2022-11-02

**Authors:** Heling Fan, Xingxiang Yan, Mingqing Fu, Difa Liu, Abdul Waheed Awan, Ping Chen, Syed Majid Rasheed, Ling Gao, Rongping Zhang

**Affiliations:** 1College of Tropical Crops, Hainan University, Haikou 570228, China; 2Tropical Crops Genetic Resources Institute (CATAS), Danzhou 571737, China; 3Department of Agriculture, Bacha Khan University, Charsadda 24461, Pakistan; 4College of Horticulture, Hainan University, Haikou 570228, China

**Keywords:** papaya, PRSV, biological agents, disease resistance, differentially expressed genes

## Abstract

The papaya industry is mainly impacted by viral diseases, especially papaya ringspot disease (PRSD) caused by papaya ringspot virus (PRSV). So far, research on the interaction between Chitosan, Lentinan and Ningnanmycin on PRSD has not been reported. This research studied the controlled and interactive effect of three biological agents, namely, Chitosan (C), Lentinan (L) and Ningnanmycin (N), on PRSV in papaya, individually and collectively. The changes in disease index, controlled effect, Peroxidase (POD), Polyphenol oxidase (PPO), Superoxide dismutase (SOD), growth and development of plants were observed at the seedling stage, in pots, and at the fruiting stage, in the field. The appearance and nutrient contents of fruits were measured during the fruit stage. The disease index of PRSV, at seedling and fruiting stages, was significantly lower for chitosan, lentinan and ningnanmycin and their interactive effect, compared to a control check treatment. The activity of the defense enzymes could be improved by the three kinds of biological agents and their interactive effect, especially lentinan and ningnanmycin. The chlorophyll content, plant height, stem diameter and fruit quality rose significantly under chitosan, lentinan and ningnanmycin treatments. The interaction of LN could inhibit PRSV disease at the seedling and fruiting stages of papaya, and promote the growth of plants and the quality of fruit at the fruit stage. Hence, this study provides the theoretical foundation for the biological control of papaya ringspot disease.

## 1. Introduction

Papaya (*Carica papaya* L.) is a popular fruit crop, native to Central America, which has high nutritious value, abundant carbohydrate, vitamins (A and C) and minerals (copper and magnesium). It is widely cultivated in tropical and sub-tropical regions of the world. Furthermore, papaya latex is widely used in the pharmaceutical industry [1]. Papaya ranks third in tropical fruit production, with a global production of 16.7 × 10^4^ tons in 2018. China is among the top global producers of papaya. However, the yield per hectare of papaya has been decreasing since 2014 [2]. Infection by a variety of viral diseases, especially PRSD caused by papaya ringspot virus (PRSV), is the prime limiting factor of papaya cultivation in countries, including China. PRSV belongs to the Potyvirus family, *Potyviridae*, and is mainly discontinuously spread by aphids [3]. PRSV has a filamentous, curved, rod shape, and a single-stranded RNA genome of about 10,300 nucleotides (nt) [4]. There is no specific bactericide to control plant viral disease [5], and the vulnerability of plants to different kinds of pathogens reduces crop yields to a great extent [6]. The best strategy to control viral infections and protect the natural environment is by inducing the plant defense response and systemic resistance [7]. Previous studies showed that Chitosan, Ningnanmycin and Lentinan could induce resistance to papaya ringspot virus (PRSV) [8].

Chitosan is a multi-cation heteropolysaccharide composed of *N*-acetyl-_D_-glucosamine and *D*-glucosamine through β-(1→4) glycosidic bonds. Despite chitosan being a natural compound (present in Zygomycetes), it is obtained principally through deacetylation of chitin, a crucial component of the fungal cell wall and arthropod exoskeleton [5]. Chitosan is a biodegradable nontoxic compound that can not only induce plants to generate systemic acquired resistance (SAR) to pathogens [9], but can also promote a defense mechanism against viral diseases, such as PRSV, cucumber mosaic virus (CMV), potato virus X (PVX) and tobacco Mosaic virus (TMV) [10,11]. The difference in molecular weight, viscosity, pKa value, degree of polymerization and degree of deacetylation of chitosan polymers have different impacts on their physicochemical and biological properties [5]. The bioactivity of chitosan can be explained by the poly-cation property of chitosan, closely related to the anion content of the target organism. More specifically, chitosan interacts with cell walls and membranes to make them unstable as an antimicrobial agent. It interferes in the transcription and translation mechanisms because of its interaction with DNA and proteins [12]. Chitosan can also chelate nutrients, trace elements and metal ions necessary for microbial growth to form a polymer film to affect the excretion of metabolites and the absorption of nutrients [9]. Lentinan, derived from the fruiting body of Lentinus edodes, is a neutral polysaccharide having three single *β*-glucose branches randomly substituted at position 6 for every five main-chain D-glucose residues. Besides having antibacterial activities, Lentinan is supposed to slow down the infectious behavior of both enveloped and non-enveloped viruses. Lentinan, in association with the tobacco Mosaic virus (TMV) coat protein and activation of a few defense genes, could also control TMV disease [13]. Ningnanmycin, a microbial pesticide with boosting resistance, high potency, and low toxicity, is isolated from Streptomyces fermentation broth. Studies have revealed that Ningnanmycin can boost the expression of disease-related proteins {phenylalanine ammonia-lyase (PAL), (POD) and (SOD)}, improve the biosynthesis of salicylic acid (SA), induce systemic resistance of host plants, and restrict the in vitro polymerization of tobacco Mosaic virus (TMV) [14,15].

Transcriptome technology is of great significance in plants to identify genes and study the differences in gene expressions for cold, drought, salinity and biotic stress resistance. PRSV is a devastating virus disease that causes plant damage due to pathogen infection. Studies on tomato yellow leaf curl virus (TYLCV) revealed that the proportion of up-regulated differentially expressed genes (DEGs) in resistant varieties of tomato was higher than that in susceptible tomato lines. The tomato defense response to TYLCV was characterized by induction and regulation involved in cell wall reorganization, transcriptional regulation, defense response, ubiquitination, metabolite synthesis and a series of gene expressions [16]. Through transcriptome sequencing technology (RNA-Seq), Chao et al. screened out the differentially expressed genes of Plutella xylostella resistance to chlorantraniliprole (R_f_) and flubendiamide (R_h_) that are enriched in pathways such as metabolic process and stress response [17]. Transcriptome sequencing of the glumes at the heading stage of mutant (M) and wild-type plants (WT) of barley showed that the formation of M may be due to the inhibition of the expression of chloroplast formation and development-related genes, resulting in the decline of photosynthesis [18,19]. In addition, through the transcriptomic analysis of the sweet potato virus infection (Sweet potato virus disease, SPVD) pathogen, it was found that SPVD infection can reduce crop photosynthetic efficiency, reduce phenylalanine metabolism pathway damage and down-regulate key enzymes, such as flavonoids, lignin, and phytochemicals [20]. High-throughput sequencing has become the main quantitative transcriptome analysis platform. It has the characteristics of low cost, high speed, and high coverage, and is widely used in the study of stress resistance of various commercial crops. At present, the whole genome sequencing of papaya has been completed, with size of 372 MB, and 28,629 genes. This has laid a good foundation for the subsequent study of the papaya transcriptome [21].

The interactive impact of Chitosan, Ningnanmycin and Lentinan, on papaya ringspot virus has not yet been studied. This pot, as well as field, study was conducted to explore the interaction effects of Chitosan, Lentinan and Ningnanmycin on PRSV, on the vegetative and reproductive growth of the plant, the PPO, POD, SOD activities in the plant leaves and gene expression in papaya. This study provides an effective means both for optimal fruit yield and lower environmental risks.

## 2. Materials and Methods

### 2.1. Test Materials and Instruments

The papaya cultivar was No.2 in Tai Nong; the PRSV pathogen was obtained from the Institute of Biotechnology, Chinese Academy of Tropical Agricultural Sciences (Haikou, China). Chitosan was chemically pure, obtained from Shandong Weifang Haizhiyuan Biological Products Co., Ltd. (Shouguang, China); Ningnamamycin, 8%, water, was obtained from Deqiang Biological Co., Ltd. (Harbin, China); Lentinan, 1%, water, was obtained from Nantong Shenyu Green Pharmaceutical Co., Ltd. (Nantong, China). The Total superoxide dismutase (T-SOD) test kit (A001-1), Peroxidase (POD) test kit (A084-3), and Polyphenol oxidase (PPO) test kit (A136-1-1) were obtained from Nanjing Jiancheng Bioengineering Institute (Nanjing, China); and the Reducing Sugar test kit (BC0235) from Beijing Solaibao Biochemical Reagent Co., Ltd. (Beijing, China). The other reagents used were of analytic grade. The Instruments used were: ULTRAVIOLET scenery photometer UV755B (Shanghai Precision Scientific Instrument Co., Ltd., Shanghai, China); Digital display constant temperature water bath (Changzhou Aohua Instrument Co., Ltd., Changzhou, China); Electronic balance (Sartorius Scientific Instruments Limited, Beijing, China); Enzyme standard instrument; Chlorophyll SPAD-502 instrument (Spectrum Technologies Ltd., Bridgend, GB); Centrifuge TGL-16MS (Shanghai Lu Xiangyi Company, Shanghai, China); JS-8600B gel imaging analyzer (Shanghai Peiqing Company, Shanghai, China); Gene amplifier HeMa9600 (Zhuhai Black Horse Company, Zhuhai, China).

### 2.2. Experiment Design

The experiment was designed in such a way that a total of 9 treatments, as shown in Table 1, were applied equally at seedling and fruiting stages. The optimal concentration of all bio-pesticides obtained from previous experiments, including chitosan 0.5 g L^−1^, Lentinan 10 g L^−1^ and Ningnanmycin 0.05 g L^−1^, was applied.

Pot experiment design: A completely randomized pot experiment was conducted on papaya seedlings at the on-campus Teaching Practice Base (20°3′39″ N 110°19′8″ E) of Haidian Campus, Hainan University, Haikou, Hainan, China. Potted plants were planted with 60% nutrient soil and 40% healthy pastoral soil. A total of 9 treatments were replicated thrice per treatment and 5 pots per replicate, making a total of 15 pots per treatment. The temperature in the greenhouse was maintained between 18~28 °C, and the humidity was kept above 60%. An amount of 200 mL of each treatment was applied as root irrigation and foliar spraying. All the treatments were inoculated with PRSV virus.

Virus inoculation method: An amount of 9 g of virus infected leaves was ground into powder with proper liquid nitrogen in a ratio of 3:100, and then 300 mL phosphoric acid buffer (pH = 7.8) was added before grinding to homogenate. Then, the residue was filtered out, and stored in a −4 °C ice box, before carrying out virus inoculation. Before virus inoculation, hands were washed with soap, and the finger dipped in 1.5 mL of inoculum (1.5 mL plant^−1^) and a small amount of quartz sand. The mesophyll tissue near the central vein of the third or fourth unfolded leaf of the plant was rubbed, which caused a slight wound on the leaf surface.

### 2.3. Investigation of Disease Index and Control Effect

Since papaya leaves experience a series of reactions after inoculation with PRSV virus, the disease was graded according to certain criteria. The individual plants were visually observed for PRSV with the following scale at regular intervals (Figure 1) [22].

0—No symptoms.

1—Having ≤3 leaves, with slightly uneven veins, slight deformity, thin new leaves, or no obvious dwarfing.

2—Having 1/4~1/3 of the leaf as malformed Mosaic, thin new leaves, yellowish leaves, stem with slight water spots.

3—Having 1/3~1/2 of the leaves deformed and Mosaic, and obvious water spots in the stem.

4—Having leaves more than 3/4 malformed Mosaic, plants thin.

5—Having leaves of the whole plant malformed, Mosaic or necrotic, and the leaves having thin chicken feet.

Disease Index and Control Effect was calculated by the following formula:(1)$$\mathrm{DI}\left(\%\right)=\frac{\sum X_{i}}{n\times \max\mathrm{rating}}\times 100$$

DI—Disease Index, ∑X_i_—Sum of all ratings, n—numbers of plants observed.
(2)$$\mathrm{CE}\left(\%\right)=\frac{{\mathrm{DI}}_{0}-{\mathrm{DI}}_{\mathrm{tr}}}{{\mathrm{DI}}_{0}}\times 100$$

CE—Control Effect, DI0—DI of control treatment, DItr—DI of biological treatment.

The virus was inoculated 1 day after treatment with bio-pesticides, and the incidence and nature of changes observed and recorded on the 1st, 3rd, 5th, 7th, 14th, 21st, 27th, 35th, 42nd and 58th days after treatment, and data with visible changes were selected for analysis.

### 2.4. Determination of Related Physiological Indexes in Papaya

After treatment with bio-pesticides, plant height, stem diameter, chlorophyll and dry matter were measured on the 40th day of the seedling stage. Plant height, stem diameter and chlorophyll were measured on day 69 of the fruiting period. The SPAD-502 chlorophyll content tester was used. The third expanded leaf of the plant was selected to measure its SPAD value.

Determination of fruit-related indicators: the fruit was inactivated at 110 °C for 30 min, and dried at 70 °C to measure its water content at constant weight. An amount of 5 g papaya fruits was added to liquid nitrogen and ground into homogenate, then centrifuged at 5000 rpm for 10 min to obtain the supernatant and measured with a hand-held refractometer. At the seedling stage and at the fruit stage, the third expanding leaves were taken at 0, 7, 14 and 21 days after treatment and stored at −20 °C [23]. The activities of POD, SOD and PPO enzymes were tested. An amount of 0.5 g papaya leaves was added to 1.5 mL phosphoric acid buffer (0.1 mol L^−1^, pH = 7.8), liquid nitrogen and quartz sand, ground into homogenate, centrifuged at 15,000 rpm at 4 °C for 20 min, and the supernatant was used as a crude enzyme solution for subsequent enzyme activity determination.

### 2.5. RNA-Seq Library Preparation and Sequencing

The symptoms of the disease appeared on the leaves of the plant. Therefore, a total of six samples of papaya leaves and leaf tissue were taken thrice. At 21 days after the onset of disease and treatment, the samples were immediately frozen in liquid nitrogen for 30 min, and then stored in dry ice. They were sent to Guangzhou Saizhe Biotechnology Co., Ltd. for transcriptome sequencing analysis. The basic process of the RNA-Seq test is divided into: (1) sample RNA fragmentation processing (2) reverse transcription synthesis cDNA library (3) DNA sequencing adapter ligation (4) PCR amplification library (5) deep sequencing (6) RNA sequence information. Preliminary detection was performed, followed by Nanodrop micro-UV spectroscopy and agarose gel electrophoresis for quantification, then followed by Agilent 2100 fragment detection (Agilent, Santa Clara, CA, USA) to comprehensively evaluate the quality of the samples. After the RNA samples were qualified, the mRNA of papaya leaves was enriched with Oligo(dT) magnetic beads. Fragmentation buffer was added to break the mRNA into short fragments. Using the mRNA as a template, a six-base random primer (Random Hexamers) was used to synthesize the first-strand of cDNA, and then buffer, dNTPs, DNA polymerase I and RNase H were added to synthesize the second-strand of cDNA. The double-stranded cDNA was purified using AMPure XP beads. The purified double-stranded cDNA was first subjected to end repair, A-tailed and ligated with sequencing adapters, and then AMPure XP beads were used for fragment size selection. Finally, PCR amplification was carried out, the PCR products were purified with AMPure XP beads to obtain the final library. Qubit 2.0 was used for preliminary quantification of the constructed library. To detect the size of the insert in the library, the library was diluted, and Agilent 2100 (Agilent, Santa Clara, CA, USA) used. After the insert reached the desired location, Q-PCR method was used to accurately quantify the effective concentration of the library in order to determine the size of the insert. Different libraries after qualification, were pooled to the flow cell, according to the requirements of effective concentration and target data volume. Following CTS-3OT clustering, the Illumina high-throughput sequencing platform (HiSeq/MiSeq) was used for sequencing.

### 2.6. RNA-Seq Data Analysis

Raw reads were filtered to remove adaptors and low-quality bases using Trimmomatic (v0.36) [24]. The trimmed and filtered reads were mapped and quantified according to the *Carica papaya ASGPBv0.4* reference genome [21], using the TopHat2 [25] aligner. Reading of quantification per transcript was performed using the Cufflinks package. In light of the different treatment methods of multiple samples, we used Cuff merge to merge results and acquire the gene expression level in different groups [26]. Finally, edgeR [27] was employed to normalize read counts and determine differentially expressed genes. Differentially expressed genes were defined as having an adjusted *p*-value < 0.05 and an absolute log2 fold change >1.0.

### 2.7. The qPCR Fluorescence Quantitative Test Method

Seven genes were randomly selected from the differentially expressed genes in CK-vs-LN, and cDNA was used as the template for qPCR detection. The sequences of qPCR primers are shown in Appendix A.

The qPCR experiment procedure: An amount of 1 μL of RNA was extracted from papaya samples and collected for reverse transcription, performed with an Eastep RT Master Mix kit from Promega (Madison, WI, USA). Total RNA synthesized the first chain of cDNA. The DNA template was extracted with specific primers for different genes (Appendix A). The qPCR test was carried out with SYBR Green dye, and genes were quantitatively analyzed. Each sample was allocated to a 20 L PCR reaction system, as shown in Appendix A. An MX3005p Real-Time PCR Thermal Cycler (Agilent Stratagene, Santa Clara, CA, USA) at 95 °C for 10 min was used for pre-denaturation, 95 °C for 15 s, 60 °C for 20 s, 72 °C for 20 s and 45 thermal cycles. Finally, the PCR program melting curve was amplified by PCR.

### 2.8. Statistical Analysis of Data

The data were analyzed by Excel and DPS 7.05, and Duncan’s new complex range method was used for multiple comparison. Drawing was by by means of GraphPad Prism 8.0.2. Origin 9.8.0.200 was used for plotting principal component analysis.

## 3. Results

### 3.1. Effects of Chitosan, Lentinan and Ningnanmycin on PRSD during Seedling Stage

#### 3.1.1. PRSD Disease Index and Control Effect during Seedling Stage

The disease index (DI) of C, N and L were significantly lower than CK treatment, while close to NT treatment. The control effect (CE) ranged between 37.02% to 96.30% on the 7th to 21st day after inoculation, the effect of interactive treatments was higher than no treatment (Figure 2). It was revealed from the pot experiment at the seedling stage, that the disease index of PRSV was significantly reduced after C, L and N treatments, and its control effect on PRSV was better after interaction.

#### 3.1.2. Changes of POD, SOD and PPO in Seedling Stage in Papaya

The activities of SOD, POD, and PPO significantly increased for C, N and L, as compared to CK. (Figure 3). C, N and L, individually and collectively, could induce plants to significantly increase resistance to PRSV, and this was consistent with DI results (Table 2). In addition, L×N had significantly positive effects on the activities of the three enzymes over a long period of time. The enzymatic activity of POD, SOD and PPO in leaves were improved to varying degrees after treatment. The PPO enzyme activity significantly improved and could maintain a high level of 108.9% to 133.3% for a long period of time, compared to CK at 21 days.

#### 3.1.3. Growth of Stem and Leaf in Seedling Stage in Papaya

The plant height, stem diameter, chlorophyll SPAD and dry weight for different treatments and their interactions were all significantly higher than NT and CK treatments. The result indicated that the three biological control agents and their interactions could promot better and stronger growth. This was consistent with DI and enzyme activities performances (Table 3). After PCA analysis, the four physical and chemical indices held relatively close ratios in principal component 1. Different treatments were divided into different regions, and LN still had a high score (Figure 4).

### 3.2. Effects of Chitosan, Lentinan and Ningnanmycin on PRSV during Fruiting Period

#### 3.2.1. PRSD Disease Index and Control Effect during Fruiting Period

The disease index (DI) of C, N and L, and their interactions, were significantly lower than check (CK) treatment at the fruiting stage. The performance of NT treatment was significantly similar to biological agents. The control effect (CE) varied between 46% and 93% on the 7th to 42nd day after disease occurred. The interactive effect of different treatments was higher than for no treatment (Figure 5). It was evident that the interactive effect of C, N and L could effectively resist PRSV at the fruiting stage as well.

#### 3.2.2. Changes of POD, SOD and PPO in Fruiting Period in Papaya

In the field experiment, the enzyme activities of POD, SOD and PPO increased to varying degrees after treatment with C, L and N. High interaction among the treatments and their interaction with different enzymes were revealed (Figure 6). The results of enzyme activity were consistent with DI and CE, and the disease index was lower in the treatment with higher enzyme activity. Interaction analysis of different enzymes revealed that Lentinan had a significant effect during the three fruiting periods (Table 4).

#### 3.2.3. Growth of Stem, Leaf and Fruit in Fruiting Stage

The plant height revealed highly significant differences for L, N and CN and was believed to promote stem growth. Similarly, L and interaction of CL revealed highly significant differences for stem diameter. The interaction effect was also significant for chlorophyll SPAD (Table 5). Comprehensive analysis showed that the strongest treatments were LN and CLN.

#### 3.2.4. Growth Analysis of Fruit in Papaya

The interaction of C, L and N promoted growth of papaya fruit, such as heavier, bigger, juicier, and sweeter fruit compared to CK. The results revealed that C, L and N were beneficial to fruit growing, and the best treatment was LN (Table 6). The principal component analyses, PC 1 and PC 2, revealed high values for LN. Similarly, fruit length, fruit diameter, fruit weight and reducing sugar had higher proportions in PC1, while in principal component 2, sweetness had a higher proportion (Figure 7).

### 3.3. Transcriptome Analysis of Leaves of Papaya

#### 3.3.1. RNA Quality Assessment

The test results were based on the DNA/RNA sample determination criteria. When RIN (RNA integrity number) was less than 1.8, the tissue fluid might contain other impurities, causing contamination to the RNA sample. When the RIN was greater than 2.2, it meant that the RNA sample had been hydrolyzed into a single nucleic acid at OD260/280, and the OD260/280 value of RNA was between 1.82 and 2.2, which was within the standard range. As shown in Appendix A, the OD260/280 values were all between 1.979 and 2.036, and the test results were placed in class A (Appendix A). This indicated that the quality of the sample met the requirements for library construction and sequencing, and the total amount met the needs of two or more library constructions.

#### 3.3.2. Illumina High-Throughput Sequencing and Comparison

Sequencing was performed using the Illumina high-throughput sequencing platform (HiSeq), and the filtered data was compared with the reference genome using TopHat software, followed by cufflinks for gene assembly. These results showed that the total reads of the six samples reached 31.635–42.409 million, covering all papaya gene sequences. The sequence contrast ranged from 74.39% to 84.48%. Part of the data is shown in Table 6.

#### 3.3.3. Analysis of 22 DEGs

Two groups of papaya were sequenced in RNA sequencing experiments, and FDR and log2FC were used to screen differential genes. The screening conditions were FDR < 0.05 and |log2FC| > 1. The abscissa of the volcano plot represents the logarithm of the fold difference between the two samples, the ordinate represents the negative Log10 value of the FDR of the two samples, the red expression is significantly up-regulated, the green is significantly down-regulated, and the blue is the criterion of FDR < 0.05. The differentially expressed genes were used to obtain a volcano plot (Figure 8). These results showed that in CK-vs-LN, a total of 1261 differentially expressed genes were involved in the gene expression of papaya after LN treatment. Of these, 900 genes were up-regulated and 361 genes were down-regulated.

#### 3.3.4. GO Analysis of DEGs

In order to further study the relationship between the gene regulation of papaya after disease incidence and bio-pesticides treatment, and the decline of the disease index, we used GOseq software to perform GO enrichment analysis of differentially expressed genes, based on NR functional annotation. The results showed that in CK-vs-LN, there were 17 differences in the biological process, among which the top 5 genes had differences in metabolic process (217 genes were up-regulated, 90 genes were down-regulated), cellular process (192 genes were up-regulated, 101 genes down-regulated), single-organism process (120 genes up-regulated, 72 genes down-regulated), response to stimulus (35 genes up-regulated, 21 genes down-regulated), and developmental process (14 genes up-regulated, 26 genes down-regulated). There were 12 differences in the cellular component, among which the top 5 genes contained differences in cell part (168 genes up-regulated, 56 genes down-regulated), cell (168 genes up-regulated, 56 genes down-regulated), membrane (155 genes up-regulated 47 genes down-regulated), organelle (128 genes, 51 genes down-regulated), and membrane part (125 genes up-regulated, 41 genes down-regulated). There were 7 differences in molecular function, including the number of differential genes before catalytic activity (176 genes up-regulated, 88 genes down-regulated), binding (135 genes up-regulated, 67 genes down-regulated), transporter activity (19 genes up-regulated, 6 genes down-regulated), nucleic acid binding transcription factor activity (7 genes were up-regulated, 3 genes were down-regulated), and electron carrier activity (2 genes were up-regulated) (Figure 9).

There were a total of 251 DEGs genes with pathway annotations, of which there were 19 pathways with significant differences (*p* < 0.05). Among them, the most significant pathway was Photosynthesis (Figure 10), in which 30 DEGs accounted for 11.95% of the above DEGs. As shown in Figure 11, after LN treatment, a total of 27 symbols were significantly up-regulated. Studies have shown that lentinan and ningnanmycin can significantly improve the photosynthesis of papaya leaves and increase the photosynthesis of crops, thereby improving the disease resistance of crops.

#### 3.3.5. The qPCR Verification of 7 Genes

This experiment focused on the relationship between LN treatment and CK. A total of 900 genes were significantly up-regulated in LN-vs-CK, and 7 genes were randomly selected, namely evm.TU.supercontig_20.79, evm.TU.supercontig_44.71, evm.TU.supercontig_42.87, evm.TU.supercontig_77.100, evm.TU.supercontig_2.212, evm.TU.supercontig_118.34, evm.TU.supercontig_19.185. The results of qPCR showed that the corresponding gene expression of papaya was up-regulated after LN treatment, which was consistent with previous experimental results (Appendix A).

## 4. Discussion

### 4.1. Control Effects of Chitosan, Lentinan and Ningnanmycin on PRSV in Papaya

In this study, the disease index of PRSV under treatment of Chitosan, Lentinan and Ningnanmycin was significantly lower than that of CK (clean water, no bio-pesticides used) at seedling and fruiting stages. Previous studies have shown that the use of chitosan for papaya PRSV virus can reduce its disease index by 10.30% [28]. Previous research has proved that chitosan’s resistance to *Botrytis cinerea* was 100% using the vitro detached leaf method with 1 mg L^−1^ N, N, N-(diethyl-*p*-dimethy laminobenzyl) chitosan [29]. Some studies have shown that the alginate-lentinan-amino-oligosaccharide hydrogel was able to induce plant resistance continuously and strongly against TMV and increase the release of calcium ions to promote *Nicotiana benthamiana* (*N. benthamiana*) growth [30]. Ningnanmycin has been used as a positive control in many new bio-pesticides, which fully demonstrate the high applicability of Ningnanmycin as a common bio-pesticide [31,32,33]. The above results were in confirmation of our findings. It was fully illustrated that Chitosan, Lentinan and Ningnanmycin, to some extent, controlled the effect of the common plant disease, but very few studies have been conducted to determine the interaction effects of a variety of biological pesticides. This study can provide a certain theoretical basis regarding several biological pesticides’ interactions. It was evident from the present study that Lentinan (10.00 g L^−1^) and Ningnanmycin (0.05 g L^−1^) can control the PRSV virus of papaya, their effects achieving 71.63~100.00% and 77.48~100.00% at seedling and fruiting stages over different time periods, respectively.

### 4.2. Defense Enzymes (POD, SOD, PPO) Regulating of Chitosan, Lentinan and Ningnanmycin in Papaya

It was revealed that the three kinds of biological control agents could increase the activity of defense enzymes. Significant interaction effect was observed when treatments were applied in combinations; however, the optimal treatment was LN. The activities of defense enzymes in plant could reach a high value at seedling and fruiting stages, hence improving resistance to external stress and injury, including PRSV. In addition, some studies have shown that *Ganoderma lucidum polysaccharide* (GLP), which is the main active molecule of *G. lucidum* used in cotton, when used in GLP spray and irrigation root treatments, can significantly increase the activities of POD, SOD and PPO in leaves, while the content of malondialdehyde decreases. After soaking in GLP, the seedling height and cotton fusarium wilt resistance both increased to some extent, and the effects were dose dependent [34]. Root rot is common for kiwifruit (*Actinidia chinensis* var. *diliciosa*) plants. Bio-agent treatments significantly improved microbial activity and changed microbial structure, increased the diversity, richness, and uniformity of microbial species, and altered the relative utilization ratio of six carbon sources. Activities of defensive-related enzymes in bio-agent treatments were significantly higher than in traditional fertilizer treatment (*p* < 0.05) [35]. When T. *harzianum* was used as a biocontrol agent, the detected hydrolytic enzyme activities of tomato were higher than with the other treatments [36]. Stem rot (*Sclerotium rolfsii* Sacc.) resistance in groundnut genotypes was due to activities of defense enzymes, such as, catalase, peroxidase, and polyphenol oxidase. Salicylic acid induced systemic resistance to enhance the activity of the defense enzymes. Enzymes catalase, peroxidase, polyphenol oxidase and chitinase showed strong negative correlation with disease severity index [37]. Therefore, we use bio-pesticides to improve the enzyme activity of crops to reduce the severity of crop diseases. Lentinan could control TMV incidence, and the action mechanism might be associated with TMV coat protein and activation of some defense genes [13]. So, through improving the activity of defense enzymes, some agents could improve the defense effect on viruses, such as PRSV. It was proved that Chitosan, Lentinan and Ningnanmycin could be used to control PRSV in the papaya industry.

### 4.3. Molecular Mechanism of Resistance to PRSV Induced by Bio-Pesticides in Papaya

To understand the cause of papaya resistance, papaya was treated with a compound biopesticide preparation LN (Lentinan 10 g L^−1^ + Ningnanmycin 0.05 g L^−1^). We found a total of 1261 differentially expressed genes, 900 up-regulated genes and 361 down-regulated genes, compared with CK treatment. For LN treatment, there were significant differences in 19 metabolic pathways, such as photosynthesis. Studies have shown that bio-pesticides improve plant disease resistance and crop growth by regulating gene expression. The pathways, such as nitrogen metabolism, carbon metabolism, carbon fixation in photosynthesis, photosynthesis and photosynthesis-antenna protein, were activated, thereby significantly promoting photosynthesis, as we hypothesized. Research by Chen Y and Venzhik Y showed that when wheat was influenced by virus and low temperature, its photosynthesis was affected [38,39]. This also indirectly proved that plant resistance was related to photosynthesis. It was in agreement with a previous study which suggested that the plant resistance was associated with photosynthesis ability [40]. In addition, other studies showed that the overexpression of genes of photosynthesis could improve resistance to the virus of maize plants [41,42,43]. Meantime, in our research, after LN treatment, a total of 27 symbols were significantly up regulated, and the resistance of papaya increased. Therefore, this study implies that the reason for increasing disease resistance of PRSD in papaya was attributed to the up-regulation of photosynthesis-related genes. Similarly, previous study has suggested that the integrated effect of tetramycin and chitosan can effectively control leaf spot disease in kiwifruit and promote CAT, POD, PPO, and SOD activities, as well as the photosynthesis ability of kiwifruit leaves, and, thereby, improve fruit quality, such as total phenolics, total flavonoids, and soluble proteins.

## 5. Conclusions

In conclusion, Chitosan, Lentinan and Ningnanmycin could increase resistance and integrally control PRSV, and the impact was consistent with promoting growth and increasing defense enzyme activities. Interaction between Lentinan and Ningnanmycin revealed more positive impact as compared to the rest. Hence the result is of great significance in controlling PRSV in the papaya industry.

## Figures and Tables

**Figure 1 molecules-27-07474-f001:**
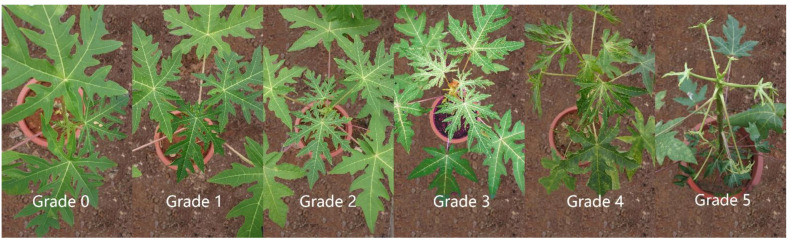
Picture of papaya seedling stage with disease index of grade 0~5.

**Figure 2 molecules-27-07474-f002:**
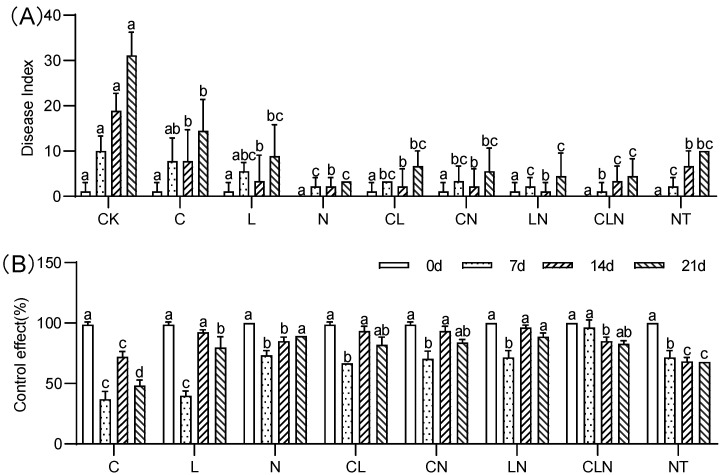
DI (**A**) and CE (**B**) of Chitosan, Lentinan and Ningnanmycin on PRSV in papaya in seedling stages. Whereas CK—control without the use of bio-pesticides, C—use of 0.50 g L^−1^ Chitosan, L—use of 10.00 g L^−1^ Lentinan, N—use of 0.05 g L^−1^ Ningnanmycin, CL—Chitosan plus Lentinan, CN—Chitosan plus Ningnanmycin, LN—Lentinan plus Ningnanmycin, CLN—Chitosan plus Lentinan plus Ningnanmycin. All the above treatments were applied by means of root irrigation with 200 mL, foliar spraying with 200 mL, and then inoculated with PRSV virus. NT—negative control without PRSV inoculation. Different lowercase letters indicate significant differences at the 0.05 level.

**Figure 3 molecules-27-07474-f003:**
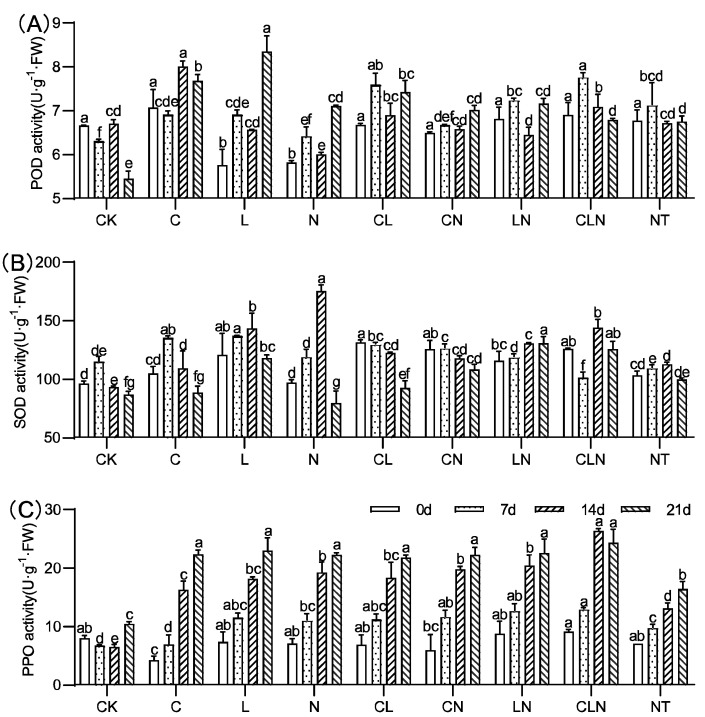
POD (**A**), SOD (**B**) and PPO (**C**) activity under treatments at seedling stage in papaya. Whereas CK—control without the use of bio-pesticides, C—use of 0.50 g L^−1^ Chitosan, L—use of 10.00 g L^−1^ Lentinan, N—use of 0.05 g L^−1^ Ningnanmycin, CL—Chitosan plus Lentinan, CN—Chitosan plus Ningnanmycin, LN—Lentinan plus Ningnanmycin, CLN—Chitosan plus Lentinan plus Ningnanmycin. All the above treatments were applied by means of root irrigation with 200 mL, foliar spraying with 200 mL, and then inoculated with PRSV virus. NT—negative control without PRSV inoculation. Different lowercase letters indicate significant differences at the 0.05 level.

**Figure 4 molecules-27-07474-f004:**
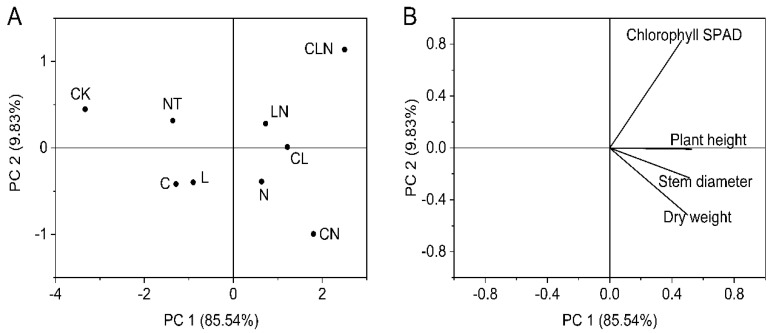
Principal component analysis of stem (**A**) and leaf growth (**B**) in seeding stage in Papaya. Whereas CK—control without the use of bio-pesticides, C—use of 0.50 g L^−1^ Chitosan, L—use of 10.00 g L^−1^ Lentinan, N—use of 0.05 g L^−1^ Ningnanmycin, CL—Chitosan plus Lentinan, CN—Chitosan plus Ningnanmycin, LN—Lentinan plus Ningnanmycin, CLN—Chitosan plus Lentinan plus Ningnanmycin. All the above treatments were applied by means of root irrigation with 200 mL, foliar spraying with 200 mL, and then inoculation with PRSV virus. NT—negative control without PRSV inoculation.

**Figure 5 molecules-27-07474-f005:**
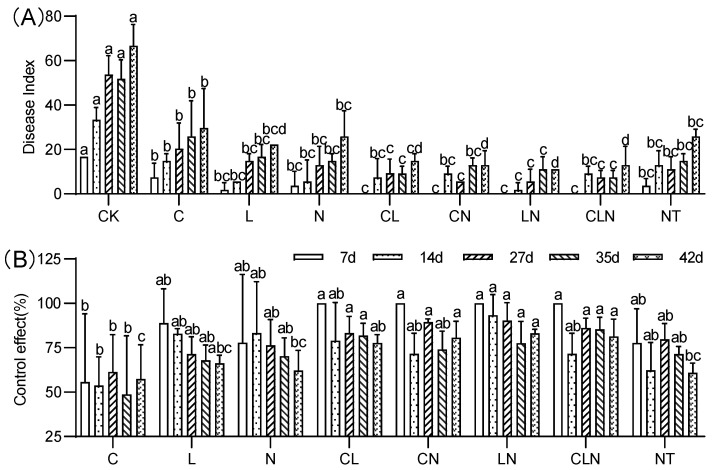
DI (**A**) and CE (**B**) of Chitosan, Lentinan and Ningnanmycin on PRSV in papaya in fruiting stage. Whereas CK—control without the use of bio-pesticides, C—use of 0.50 g L^−1^ Chitosan, L—use of 10.00 g L^−1^ Lentinan, N—use of 0.05 g L^−1^ Ningnanmycin, CL—Chitosan plus Lentinan, CN—Chitosan plus Ningnanmycin, LN—Lentinan plus Ningnanmycin, CLN—Chitosan plus Lentinan plus Ningnanmycin. Each papaya plant was irrigated with 5 L and sprayed with 6 L biological pesticide once every 7 to 14 days. NT—negative control without PRSV inoculation. Different lowercase letters indicate significant differences at the 0.05 level.

**Figure 6 molecules-27-07474-f006:**
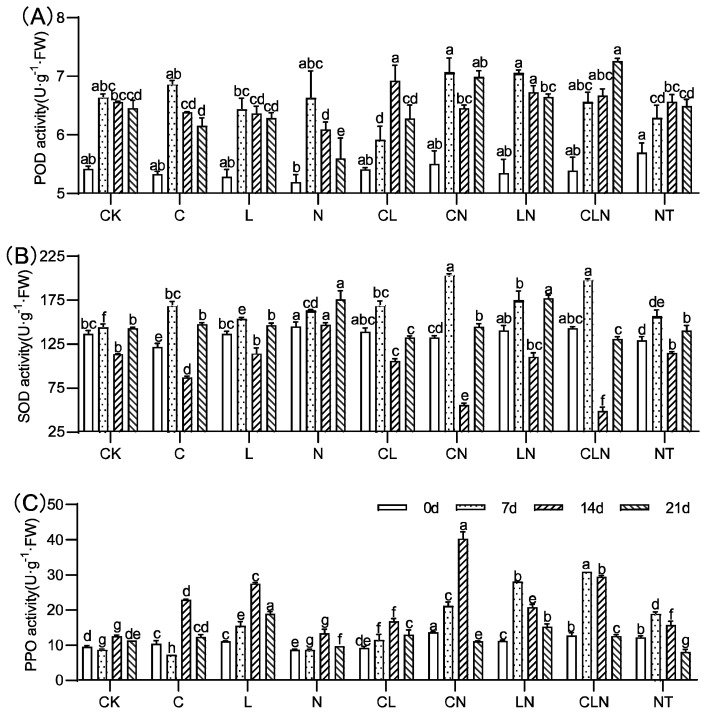
POD (**A**), SOD (**B**) and PPO (**C**) activity under treatments at the fruiting period in papaya. Whereas CK—control without the use of bio-pesticides, C—use of 0.50 g L^−1^ Chitosan, L—use of 10.00 g L^−1^ Lentinan, N—use of 0.05 g L^−1^ Ningnanmycin, CL—Chitosan plus Lentinan, CN—Chitosan plus Ningnanmycin, LN—Lentinan plus Ningnanmycin, CLN—Chitosan plus Lentinan plus Ningnanmycin. Each papaya plant was irrigated with 5 L and sprayed with 6 L biological pesticide once every 7 to 14 days. NT—negative control without PRSV inoculation. Different lowercase letters indicate significant differences at the 0.05 level.

**Figure 7 molecules-27-07474-f007:**
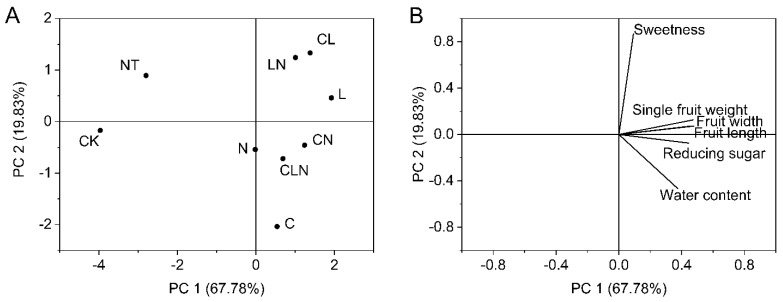
(**A**,**B**) Principal component analysis of fruit indices in the fruit stage of papaya. Whereas CK—the control without the use of bio-pesticides, C—use of 0.50 g L^−1^ Chitosan, L—use of 10.00 g L^−1^ Lentinan, N—use of 0.05 g L^−1^ Ningnanmycin, CL—Chitosan plus Lentinan, CN—Chitosan plus Ningnanmycin, LN—Lentinan plus Ningnanmycin, CLN—Chitosan plus Lentinan plus Ningnanmycin. Each papaya plant was irrigated with 5 L and sprayed with 6 L biological pesticide, once every 7 to 14 days. NT was a negative control without PRSV inoculation.

**Figure 8 molecules-27-07474-f008:**
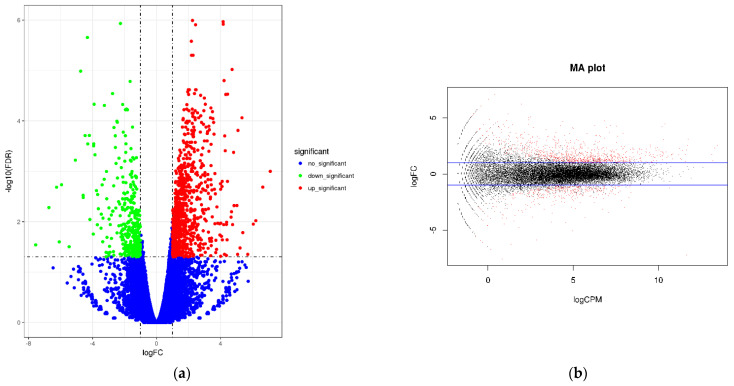
(**a**) Volcano map of gene expression. (**b**) Smear map of gene expression.

**Figure 9 molecules-27-07474-f009:**
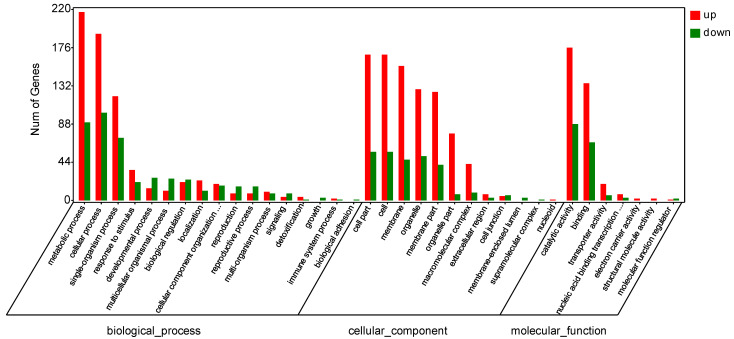
Classification of differentially expressed genes (DEGs) by gene ontology (GO) analysis: CK vs. LN.

**Figure 10 molecules-27-07474-f010:**
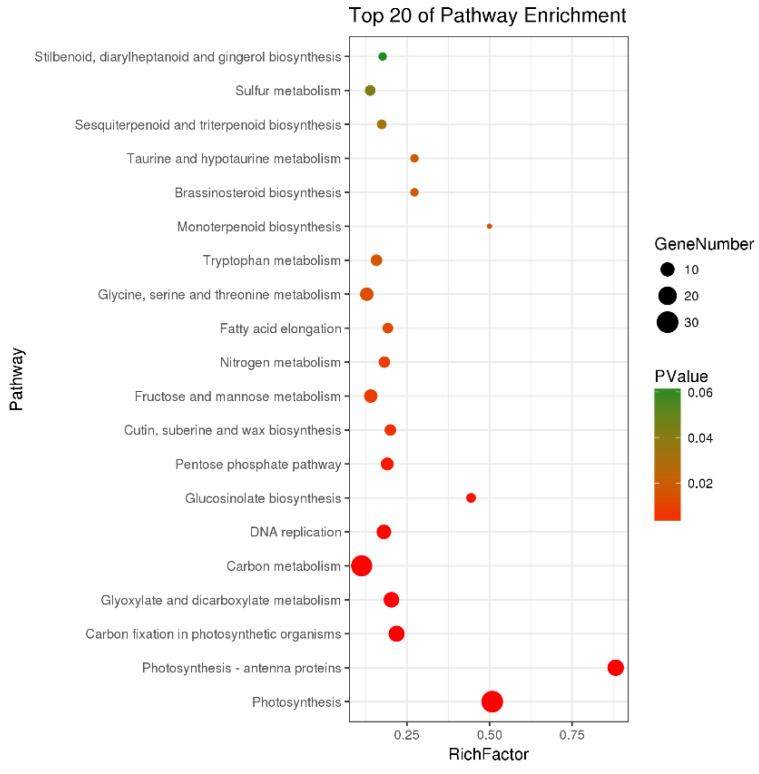
Pathway enrichment bubble map of differentially expressed genes (DEGs).

**Figure 11 molecules-27-07474-f011:**
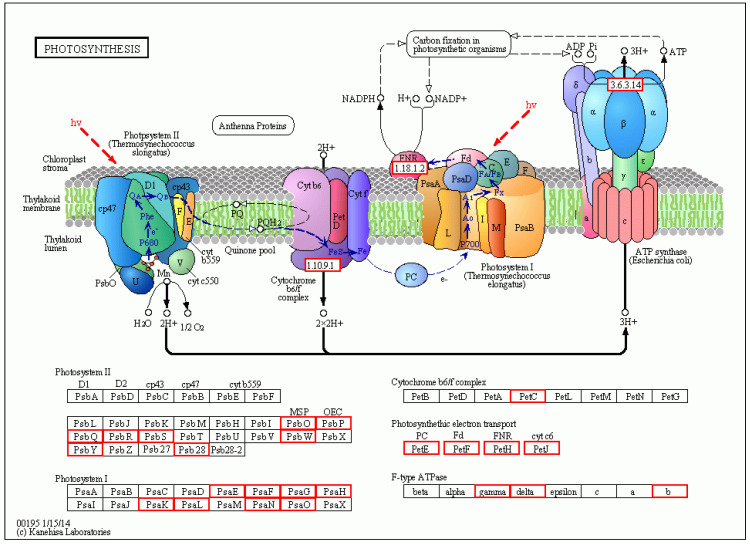
Diagram of the metabolic pathway of photosynthesis in CK-vs-LN. Red indicates related genes up-regulated in LN treatment.

**Table 1 molecules-27-07474-t001:** The experiment design of controlled effect of Chitosan, Lentinan, Ningnanmycin on PRSV (g L^−1^) in the above treatment.

Abbreviations	Chitosan	Lentinan	Ningnanmycin	Note
CK	0	0	0	inoculated with PRSV
C	0.5	0	0	
L	0	10	0	
N	0	0	0.05	
CL	0.5	10	0	
CN	0.5	0	0.05	
LN	0	10	0.05	
CLN	0.5	10	0.05	

CK—control check, C—Chitosan, L—Lentinan, N—Ningnanmycin, CL—Chitosan + Lentinan, CN—Chitosan + Ningnanmycin, LN—Lentinan + Ningnanmycin, CLN—Chitosan + Lentinan + Ningnanmycin, NT—Negative Control, no PRSV.

**Table 2 molecules-27-07474-t002:** Effect of C, N and L on POD, SOD and PPO activity during Pot and Field experiment.

Factors	Pot Experiment									Field Experiment								
	POD			SOD			PPO			POD			SOD			PPO		
	7th	14th	21st	7th	14th	21st	7th	14th	21st	7th	14th	21st	7th	14th	21st	7th	14th	21st
C×	***	***	Ns	Ns	**	Ns	Ns	***	***	Ns	*	**	***	***	***	***	***	***
L×	***	Ns	***	Ns	**	***	***	***	***	*	**	**	*	***	**	***	**	***
N×	Ns	***	Ns	***	***	***	***	***	***	*	Ns	**	***	***	***	***	***	***
C×L	Ns	*	***	***	*	***	Ns	Ns	***	**	Ns	Ns	**	***	***	***	***	***
C×N	Ns	Ns	**	**	**	***	Ns	Ns	**	Ns	Ns	***	**	***	***	***	***	**
L×N	*	***	***	***	***	**	**	*	**	*	Ns	**	Ns	***	Ns	***	***	Ns
C×L×N	Ns	*	***	Ns	***	Ns	Ns	***	***	Ns	**	*	Ns	*	Ns	***	Ns	**

Whereas CK—control without the use of bio-pesticides, C—use of 0.50 g L^−1^ Chitosan, L—use of 10.00 g L^−1^ Lentinan, N—use of 0.05 g L^−1^ Ningnanmycin, CL—Chitosan plus Lentinan, CN—Chitosan plus Ningnanmycin, LN—Lentinan plus Ningnanmycin, CLN—Chitosan plus Lentinan plus Ningnanmycin. All the above treatments were applied by means of root irrigation with 200 mL, foliar spraying with 200 mL, and then inoculated with PRSV virus. *—*p* < 0.05, **—*p* < 0.01, ***—*p* < 0.001, Ns—not significant.

**Table 3 molecules-27-07474-t003:** Growth analysis of stem and leaf in seedling stage in Papaya.

Treatments	Growth and Development Index			
	Plant Height (m plant^−1^)	Stem Diameter (cm plant^−1^)	Chlorophyll SPAD	Dry Weight (g plant^−1^)
CK	19.48 ± 2.25 ^e^	5.03 ± 0.32 ^d^	9.44 ± 0.89 ^e^	1.59 ± 0.11 ^e^
C	22.80 ± 3.71 ^de^	8.86 ± 0.43 ^c^	10.72 ± 0.97 ^e^	3.25 ± 0.48 ^cd^
L	24.41 ± 2.91 ^cde^	9.45 ± 1.39 ^bc^	11.32 ± 1.83 ^de^	3.28 ± 0.09 ^cd^
N	26.92 ± 1.42 ^bcd^	10.21 ± 0.29 ^bc^	15.98 ± 1.34 ^bc^	4.99 ± 1.06 ^ab^
CL	28.54 ± 3.58 ^abc^	11.07 ± 0.55 ^ab^	18.67 ± 2.11 ^b^	4.85 ± 0.79 ^ab^
CN	32.38 ± 4.15 ^ab^	11.99 ± 0.31 ^a^	14.97 ± 1.25 ^c^	5.59 ± 0.32 ^a^
LN	28.71 ± 3.23 ^abc^	10.12 ± 0.82 ^bc^	18.21 ± 1.49 ^b^	4.23 ± 0.02 ^bc^
CLN	33.21 ± 2.77 ^a^	11.94 ± 1.87 ^a^	25.67 ± 1.46 ^a^	4.64 ± 0.91 ^ab^
NT	21.29 ± 1.31 ^e^	9.46 ± 0.57 ^bc^	13.58 ± 2.22 ^cd^	2.42 ± 0.32 ^de^
**ANOVA**				
C×	**	***	***	***
L×	*	***	***	Ns
N×	***	***	***	***
C×L	Ns	Ns	***	Ns
C×N	Ns	Ns	Ns	*
L×N	Ns	***	Ns	***
C×L×N	Ns	Ns	Ns	Ns

CK—the control without the use of bio-pesticides, C—use of 0.50 g L^−1^ Chitosan, L—use of 10.00 g L^−1^ Lentinan, N—use of 0.05 g L^−1^ Ningnanmycin, CL—Chitosan plus Lentinan, CN—Chitosan plus Ningnanmycin, LN—Lentinan plus Ningnanmycin, CLN—Chitosan plus Lentinan plus Ningnanmycin. All the above treatments were applied by means of root irrigation with 200 mL, foliar spraying with 200 mL, and then inoculated with PRSV virus. NT—negative control without PRSV inoculation. Different lowercase letters indicate significant differences at the 0.05 level. *—*p* < 0.05, **—*p* < 0.01, ***—*p* < 0.001, Ns—Not significant.

**Table 4 molecules-27-07474-t004:** Growth analysis of stem and leaf in the fruiting stage in papaya.

Treatments	Plant Height (m plant^−1^)	Stem Diameter (cm plant^−1^)	Chlorophyll SPAD
CK	2.47 ± 0.06 ^f^	36.67 ± 4.36 ^bc^	9.42 ± 1.45 ^f^
C	2.52 ± 0.06 ^e^	33.33 ± 3.51 ^d^	15.01 ± 1.51 ^c^
L	2.64 ± 0.05 ^cd^	36.00 ± 1.00 ^bcd^	13.64 ± 2.61 ^d^
N	2.73 ± 0.05 ^b^	38.33 ± 2.08 ^abc^	15.59 ± 2.96 ^c^
CL	2.67 ± 0.06 ^c^	40.67 ± 4.51 ^a^	19.71 ± 2.73 ^b^
CN	2.62 ± 0.10 ^d^	35.33 ± 0.58 ^cd^	24.15 ± 1.43 ^a^
LN	2.84 ± 0.03 ^a^	38.67 ± 0.58 ^ab^	19.84 ± 2.94 ^b^
CLN	2.86 ± 0.09 ^a^	41.00 ± 1.73 ^a^	20.23 ± 1.54 ^b^
NT	2.53 ± 0.08 ^e^	35.33 ± 3.79 ^cd^	11.82 ± 0.98 ^e^
**ANOVA**			
C×	Ns	Ns	***
L×	***	***	***
N×	***	*	***
C×L	**	***	***
C×N	***	Ns	*
L×N	Ns	Ns	***
C×L×N	**	Ns	***

CK—the control without the use of bio-pesticides, C—use of 0.50 g L^−1^ Chitosan, L—use of 10.00 g L^−1^ Lentinan, N—use of 0.05 g L^−1^ Ningnanmycin, CL—Chitosan plus Lentinan, CN—Chitosan plus Ningnanmycin, LN—Lentinan plus Ningnanmycin, CLN—Chitosan plus Lentinan plus Ningnanmycin. All the above treatments were applied by means of root irrigation with 200 mL, foliar spraying with 200 mL, and then inoculation with PRSV virus. NT was a negative control without PRSV inoculation. Each papaya plant was irrigated with 5 L and sprayed with 6 L biological pesticide, once every 7 to 14 days. NT was a negative control without PRSV inoculation. Different lowercase letters indicate significant differences at the 0.05 level. *—*p* < 0.05, **—*p* < 0.01, ***—*p* < 0.001, Ns—Not significant.

**Table 5 molecules-27-07474-t005:** Growth analysis of fruit under treatments in fruiting stage in papaya.

Treatments	Single Fruit Weight (kg per fruit^−1^)	Fruit Length (cm per fruit^−1^)	Fruit Width (cm per fruit^−1^)	Water Content (%)	Reducing Sugar (mg g^−1^)	Sweetness
CK	0.76 ± 0.15 ^d^	22.00 ± 1.00 ^b^	23.33 ± 4.93 ^c^	85.82 ± 1.19 ^d^	3.57 ± 0.02 ^e^	10.76 ± 0.29 ^e^
C	1.18 ± 0.20 ^abc^	24.00 ± 1.00 ^a^	39.00 ± 1.73 ^ab^	90.42 ± 0.55 ^ab^	3.81 ± 0.02 ^b^	6.43 ± 0.20 ^h^
L	1.34 ± 0.21 ^a^	25.00 ± 0.48 ^a^	41.67 ± 5.77 ^ab^	88.85 ± 0.05 ^c^	3.94 ± 0.01 ^a^	13.33 ± 0.10 ^c^
N	1.04 ± 0.03 ^bc^	24.33 ± 1.15 ^a^	36.67 ± 2.89 ^ab^	90.72 ± 0.61 ^a^	3.71 ± 0.01 ^d^	11.83 ± 0.17 ^d^
CL	1.31 ± 0.09 ^a^	25.00 ± 0.67 ^a^	43.33 ± 1.53 ^a^	88.61 ± 0.66 ^c^	3.77 ± 0.03 ^c^	15.72 ± 0.25 ^a^
CN	1.29 ± 0.12 ^a^	25.00 ± 0.85 ^a^	41.33 ± 3.06 ^ab^	89.10 ± 1.25 ^c^	3.81 ± 0.01 ^b^	10.26 ± 0.29 ^f^
LN	1.27 ± 0.04 ^ab^	25.33 ± 1.15 ^a^	39.00 ± 4.58 ^ab^	88.28 ± 1.82 ^c^	3.77 ± 0.02 ^c^	15.38 ± 0.13 ^b^
CLN	1.26 ± 0.13 ^ab^	24.67 ± 0.58 ^a^	42.00 ± 4.58 ^ab^	89.33 ± 1.92 ^bc^	3.70 ± 0.04 ^d^	9.43 ± 0.01 ^g^
NT	0.99 ± 0.11 ^c^	22.00 ± 0.58 ^b^	35.00 ± 2.65 ^b^	84.08 ± 1.99 ^e^	3.57 ± 0.03 ^e^	12.10 ± 0.58 ^h^
**ANOVA**						
C×	**	Ns	**	**	***	***
L×	***	**	**	Ns	***	***
N×	Ns	*	Ns	**	***	***
C×L	**	*	*	Ns	***	***
C×N	Ns	Ns	Ns	***	*	***
L×N	*	*	*	**	***	***
C×L×N	Ns	Ns	Ns	***	***	***

CK—the control without the use of bio-pesticides, C—use of 0.50 g L^−1^ Chitosan, L—use of 10.00 g L^−1^ Lentinan, N—use of 0.05 g L^−1^ Ningnanmycin, CL—Chitosan plus Lentinan, CN—Chitosan plus Ningnanmycin, LN—Lentinan plus Ningnanmycin, CLN—Chitosan plus Lentinan plus Ningnanmycin. All the above treatments were applied by means of root irrigation with 200 mL, foliar spraying with 200 mL, and then inoculatied with PRSV virus. NT was a negative control without PRSV inoculation. Different lowercase letters indicate significant differences at the 0.05 level. *—*p* < 0.05, **—*p* < 0.01, ***—*p* < 0.001, Ns—Not significant.

**Table 6 molecules-27-07474-t006:** Statistical table of comparisons with the reference genome after filtering rRNA.

Sample	Total Pair Reads	Unmapped Pair Reads	Unique Mapped Pair Reads	Multiple Mapped Pair Reads	Mapping Ratio
CK1	31,635,080	6,301,472 (19.92%)	25,237,526 (79.78%)	96,082 (0.30%)	80.08%
CK2	36,613,018	9,377,139 (25.61%)	27,134,133 (74.11%)	101,746 (0.28%)	74.39%
CK3	34,442,498	6,132,542 (17.81%)	28,205,012 (81.89%)	104,944 (0.30%)	82.19%
LN_1	37,993,586	6,308,417 (16.60%)	31,559,478 (83.07%)	125,691 (0.33%)	83.40%
LN_2	35,965,986	5,583,588 (15.52%)	30,250,979 (84.11%)	131,419 (0.37%)	84.48%
LN_3	42,409,640	6,849,230 (16.15%)	35,368,850 (83.40%)	191,560 (0.45%)	83.85%

## Data Availability

The data presented in this study are available upon request from the corresponding author. The data are not publicly available due to privacy concerns.

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
