# Peer review of "Interactive Effect of Biological Agents Chitosan, Lentinan and Ningnanmycin on Papaya Ringspot Virus Resistance in Papaya (Carica papaya L.)"

_molecules, 2022, doi:10.3390/molecules27217474_

Round 1

Reviewer 1 Report

In this manuscript, Fan H and colleagues successfully show that the three different kinds of biological agents, individually as well as in combination can control the papaya ring spot disease. Here, the authors studied the impact of the biological compounds on the plant growth, disease progression and the growth of fruit. After having characterized the significant effect of the biological compounds Chitosan, Lentinan, Ningnanmycin and combination of these compounds. Further, they have performed RNAseq experiments followed by Differential expression analysis revealing several upregulated and downregulated genes that regulate biological processes including metabolism. The pathway analysis of DEGs showed enrichment in genes responsible for photosynthesis.

They propose that Lentinan and Ningnanmycin can control papaya ringspot virus and this impact on improving the defense mechanism. This work shows interesting findings as a detailed characterization and impact of biological compounds in improving papaya plant resistance against papaya ringspot virus disease.

We suggest a minor revision of the manuscript with the following comments addressed. 

Minor:

Line 72 TMV

Line 91 AL?

Table 1: CLN should contain Lentinan 10.

Figure 1: Chlorotic spots and mosaic symptoms could be shown with arrows or arrow heads.

Line 186 “papaya leave and leaf tissue and leaves” choice of these samples could be explained?

Line 198 and 199: Replace one-strand and two-strand with first-strand and second-strand.

Line 204-211: The sentence is isolated from being passive to active voice.

Line 228: “1 μL of RNA” the concentration of RNA would be better representation.

Line 230-231: “Total RNA or Poly (A+) RNA to synthesize first chain cDNA” unclear sentence.

Line 257: “Different lowercase letters indicate significant differences at the 0.05 level.” It be informative to explain what each letter denotes

Line 433: “Figure X”?

Line 455: “vitro mycelium” in vitro?

Line 464: “The resistance was 54~57%” ~54-57%? same in line 472.

Line 486-489: Busy sentence, could be simplified.

Line 506: “peroxidase (E1.11.1.7) genes 506 and peroxidase genes” what is the importance of mentioning twice? Is there a difference?

Line 510-516: Complex sentence, could be simplified.

Reviewer 2 Report

Dear Authors,

This paper study the Effect of Chitosan, Lentinan and Ningnanmycin on Papaya Ringspot Virus Resistance in Papaya (Carica papaya L.). It is an interesting paper but some issues must be attended to before being considered for publication,

Abstract.

Must include more quantitative data.

Introduction section.

Change the capital letters from Chitosan, Lentinan and Ningnanmycin along text if they are not at the beginning of sentences.

Line 67 .... Lentinusedodes.....check the name.

All plant scientific names must be in Italics 

Methodology section

Do you know the chitosan degree of acetylation and molecular weight?

Line 139...a total of 15 pots per treatment or total assay?

Figure 1. I suggest showing individual leaves infected with a virus, instead of plants. In general, no differences are observed among Grades.

Line 167...what do you mean by.... data with obvious changes was selected for analysis?....

Line 172... describe the SPAD chlorophyll asssay...number of samples per treatment? Statistical analysis?.

Line 180... All enzymatic analyses were performed in Phosphoric acid buffer?. Describe the methodology for each assay. They are not obvious.

Line 186...check the grammar of this sentence.

Line 185... It is your own methodology or it was previously adapted? consider adding references to all gene analyses.

Results section.

How do you select those concentrations for each product?

Figure 9 must be improved...letters are too small. ...consider to add 3 independent figures (A-C)

Line 453 to 460...The discussion is focused on chitosan effect on fungal strains control. But this is not related to this work. Consider discussing chitosan antiviral action....there are several papers on this topic.

Lentinan is a beta glucan...consider doing the same with this polymer...

Line 474.... Consider discussing the rol of these enzymes mainly in plant antiviral treatments studies....rather than antifungal.

Line 480...Ganoderma polysaccharide is a beta glucan?...if yes, just put this on the manuscript.

Line 503...the molecular mechanism is poorly discussed...

Improve discussion about both plant and fruit physiological parameters.

Conclusions must be improved.

Supplementary figure S4. Which genes are the one presented in this figure?.

Round 2

Reviewer 2 Report

No comments